# Structural insights into SSNA1 self-assembly and its microtubule binding for centriole maintenance

Lorenzo Agostini [1,6], Jason A. Pfister [2,6], Nirakar Basnet[1,5], Jienyu Ding[1], Rui Zhang [3], Christian Biertümpfel [1] ✉, Kevin F. O'Connell [2] ✉ & Naoko Mizuno [1,4] ✉

SSNA1 is a fibrillar protein involved in dynamic microtubule remodeling, including nucleation, co-polymerization, and microtubule branching. The underlying molecular mechanism has remained unclear due to a lack of structural information. Here, we determine the cryo-EM structure of *C.elegans* SSNA-1 at 4.55-Å resolution and evaluate its role in embryonic development. We find that SSNA-1 forms an anti-parallel coiled-coil, with self-assembly facilitated by an overhang of 16 C-terminal residues that form a triple-stranded helical junction. The microtubule-binding region is within the triple-stranded junction, suggesting that self-assembly of SSNA-1 creates hubs for effective microtubule interaction. Genetical analysis elucidates that SSNA-1 deletion significantly reduces embryonic viability, and causes multipolar spindles during cell division. Interestingly, impairing SSNA-1 self-assembly has a comparable effect on embryonic viability as the knockout strain. Our study provides molecular insights into SSNA-1's self-assembly and its role in microtubule binding and cell division regulation through centriole stability.

SSNA1 is a microtubule-associated protein that localizes at various subcellular regions, including centrosomes, basal bodies, the midbody of dividing cells, and axon branching points[1–7]. Although its precise role has been enigmatic, SSNA1 is essential for cell integrity, and it is highly conserved across eukaryotic species from algae to humans[7]. The suppression of SSNA1 has been shown to result in the formation of multinucleate cells and a reduction in the number of mitotic cells[3], highlighting its critical role during cell division. SSNA1 has also been identified as a factor in the regulation of axon branching[4]. The localization of SSNA1 in areas of dynamic cytoskeletal activities in processes such as microtubule nucleation, polymerization, and stabilization suggests that SSNA1 is involved in rapid cytoskeletal remodeling.

In vitro studies have characterized SSNA1 as a microtubule-binding protein that facilitates the nucleation of microtubules[4,5]. Although SSNA1 is only a small 14 kDa protein mostly comprised of a coiled-coil domain, it grows into long fibrils through a self-assembly process[4,6,7]. These fibrils can further laterally associate to form sheet-like, or bundle-like assemblies, or co-polymerize with tubulin to decorate microtubules along their axis[4]. Interestingly, SSNA1 has been shown to induce a microtubule branching in vitro, in which protofilaments of a microtubule split into several microtubules[4]. While these phenomena have been reported, the precise molecular mechanism of SSNA1's self-assembly and its biological relevance remains elusive due to a lack of structural and functional information.

[1]Laboratory of Structural Cell Biology, National Heart, Lung, and Blood Institute, National Institutes of Health, 50 South Dr., Bethesda, MD 20892, USA. [2]Laboratory of Biochemistry and Genetics, National Institute of Diabetes and Digestive and Kidney Diseases, National Institutes of Health, 8 Center Dr., Bethesda, MD 20892, USA. [3]Department of Biochemistry and Molecular Biophysics, Washington University in St. Louis, School of Medicine, St. Louis, MO, USA. [4]National Institute of Arthritis and Musculoskeletal and Skin Diseases, National Institutes of Health, 50 South Dr., Bethesda, MD 20892, USA. [5]Present address: Institute for Protein Innovation, Boston, MA 02115, USA. [6]These authors contributed equally: Lorenzo Agostini, Jason A. Pfister. ✉e-mail: christian.biertuempfel@nih.gov; kevino@intra.niddk.nih.gov; naoko.mizuno@nih.gov

The role of SSNA1 in centriole and centrosome functions is particularly intriguing. The centrosome is the primary microtubule organizing center (MTOC) in eukaryotic cells. It is essential for cell division, cell shape formation, and motility. It comprises two centrioles, which are perpendicularly oriented tube-like structures consisting of microtubule arrays around a cartwheel scaffold. The surrounding area is made of the pericentriolar material (PCM), which is less well-structured and more dynamic[8,9]. As part of the spindle apparatus the centrosome is tightly integrated into cell cycle and checkpoint control[10]. The precise duplication of each centriole during S-phase ensures that exactly two centrosomes form, directing spindle pole formation and chromosome segregation during mitosis. Proper coordination of centriole replication within the cell cycle is critical, as miscoordination can lead to cancer and ciliopathies[11,12]. To facilitate that, a conserved set of core centriolar assembly factors including the polo-like kinase ZYG-1[13], and coiled-coil proteins like SPD-2[14,15], SAS-4[16,17], SAS-5[18], and SAS-6[19,20] orchestrate the assembly of the cartwheel, the scaffold for the assembly of centrioles[21,22]. While centrioles are robust structures, the PCM is a dynamic, membrane-less organization, in which proteins condense into a micron-sized space[23-29] with centriolar proteins also contributing[10]. Interestingly, many centrosomal proteins, characterized by coiled-coil motifs, oligomerize to build higher-order assemblies[28-35], which are crucial recruiting factors required to construct the mitotic machinery. Due to its colocalization at the centrosome and microtubule-remodeling abilities, a hypothesis arises that the coiled-coil protein SSNA1 plays an active role in the maintenance of centrioles and centrosomes.

In this study, we use in vitro reconstitution and cryo-EM to obtain structural information of *C. elegans* SSNA-1 fibrils and gain insights into its molecular mechanisms. We further validate the impact of SSNA-1 in *C. elegans* through in vivo analysis. Our study reveals that the coiled-coil of SSNA-1 is staggered in an anti-parallel fashion with overhangs forming a triple-stranded helical junction and assemble into an eight-fold fibril structure. The microtubule-binding region including the key residue R18 is part of the triple-stranded junctions, indicating that the junctions act as repeating hub for effective microtubule-binding upon fibril formation. Finally, we find that SSNA-1 is associated with embryonic centrioles using *C. elegans*. Using CRISPR-mediated genome editing, we observe that deletion of the *C. elegans ssna-1* gene results in a loss of embryonic viability marked by cell division defects, multipolar mitotic spindles and extra centrosomes. Mutations that affect the ability of SSNA-1 to oligomerize or bind microtubules also resulted in a loss of embryonic viability indicating that SSNA-1 functions as a microtubule-binding fibril to control centrosome homeostasis.

## Results

### *C. elegans* SSNA-1 is required for embryonic cell division

To conduct an in-depth structural and functional study of SSNA1 during development of a multicellular organism, we used *C. elegans* as a genetically tractable model system. *C. elegans* possesses the uncharacterized gene (T07A9.13) encoding an SSNA1 homolog[36,37]. Hereafter we refer this gene as *ssna-1* and its protein product as SSNA-1.

To analyze the role of SSNA-1 in vivo, we used CRISPR-Cas9 genome engineering to precisely delete the *ssna-1* open reading frame, thereby creating the null allele *ssna-1(bs182)*, which we abbreviate as *ssna-1(Δ)* (Fig. 1A). Homozygotes containing the deletion were viable but exhibited several developmental defects (also in ref. 37). Foremost among these is an embryonic lethal phenotype; only 30.5% of the progeny of *ssna-1(Δ)* hermaphrodites survived embryogenesis (Fig. 1B, C). Homozygous animals occasionally exhibited body morphology defects such as vulva defects and body length anomalies (also in ref. 37). Overall, these data indicate that SSNA-1 plays an essential role during one or more stages of *C. elegans* development.

We next addressed the underlying cause of the embryonic lethal phenotype. Taking advantage of the optically-transparent embryo with its large nonmotile blastomeres, we imaged the early divisions of live wild-type and *ssna-1(Δ)* embryos that express GFP::histone (chromosomes), mCherry::β-tubulin (microtubules), and GFP::SPD-2 (centrosomes) (Fig. 1D). While wild-type embryos divided with bipolar mitotic spindles, we found that *ssna-1(Δ)* embryos possessed cells with multipolar spindles, multiple centrosomes, and multiple nuclei, a phenotype indicative of a cell division defect (also in ref. 37), which is consistent with the defects observed when SSNA1 is depleted in human cell lines and *C. reinhardtii*[2,3].

We then investigated the localization of SSNA-1 within the cells of the early embryo. Using CRISPR-Cas9 genome engineering, we modified the endogenous *ssna-1* gene so that it contained a four-amino-acid C-terminal EPEA epitope tag or C-tag. Animals homozygous for the *ssna-1::C-tag* gene were wild-type in both appearance and embryonic viability (Fig. 1B), indicating that the C-tag did not interfere with the function of SSNA-1. By immunostaining we found that SSNA-1::C-tag localized to nuclear associated foci that coincided with ZYG-1::SPOT, a marker of centrioles (Fig. 1E). No staining was observed in embryos that lacked the C-tag, revealing that the centrosome staining observed in the SSNA-1::C-tag expressing embryos was specific. We conclude that *C. elegans* SSNA-1, like its vertebrate ortholog, localizes to centrosomes and is required for cell division. As shown in ref. 37, the cell division defects arise from structurally compromised centrioles which fragment leading to multipolar spindles[37].

### SSNA-1 self-assembly is driven by coiled-coil connections through triple-stranded helical junctions

To gain insights into the SSNA-1 functional-structural mechanism, we recombinantly prepared *C. elegans* SSNA-1 using *E.coli* and characterized its behaviour biochemically and biophysically (Supplementary Fig. 1). SSNA1 forms a polymer made of thin fibrils[4,6,7], which can interact with microtubules, but the mechanism of assembly and its functional implication were unclear. Purified *C. elegans* SSNA-1 bound to microtubules, and also formed bundled filaments. These characteristics including microtubule binding property were independent of the purification tags (i.e. His tag, Supplementary Fig. 1C, D). However, these filaments were not suitable for structural analysis. To obtain a structurally amenable form of SSNA-1, we introduced a series of point mutations based on our previous study[4] and identified a mutant variant, SSNA-1(3E) with R18E/R20E/Q98E, suitable for further structure determination (Supplementary Fig. 2). Cryo-EM analysis of SSNA-1(3E) revealed the presence of 8 parallel fibrils assembled into filaments with C8 symmetry with a periodicity of 112 Å, which was also validated by the averaged power spectrum of the aligned filaments (Figs. 2A, B and S2C). The outer and inner diameters of the filaments were 90 Å and 55 Å, respectively (Fig. 2B). Within each fibril of the filament, individual protomers were arranged in double-stranded coiled-coils (Fig. 2C, D) and longitudinally connected to adjacent coiled-coil units via triple-stranded helical junctions (Fig. 2C–E). The cryo-EM analysis also revealed a density within the inner lumen of the filament (Fig. 2B) that hold the eight fibrils into a filament. The fibrils were laterally connected through an interfibrous density bridging terminal parts of different fibrils (Fig. 2E).

Despite the moderate resolution of our 3D reconstruction at 4.55 Å (Supplementary Fig. 2A, B, Supplementary Table 1) due to the inherent flexibility of SSNA-1 coiled-coil fibrils, we successfully generated an AlphaFold structural prediction as an initial model (Supplementary Fig. 3) and refined a hybrid structural model (Fig. 3A–F). This validated the presence of the triple-stranded helical junction. Our model building revealed that the coiled-coil within each protomer unit (CC1-a1 and CC1-a2) assembles in an antiparallel manner (Fig. 3A), spanning residues 9-105, unlike the previous publications following a conventional assumption that the coiled-coil running in parallel with little structural information[4,6,7]. The density preceding residue 9 extended and connected to the inner lumen density (Supplementary

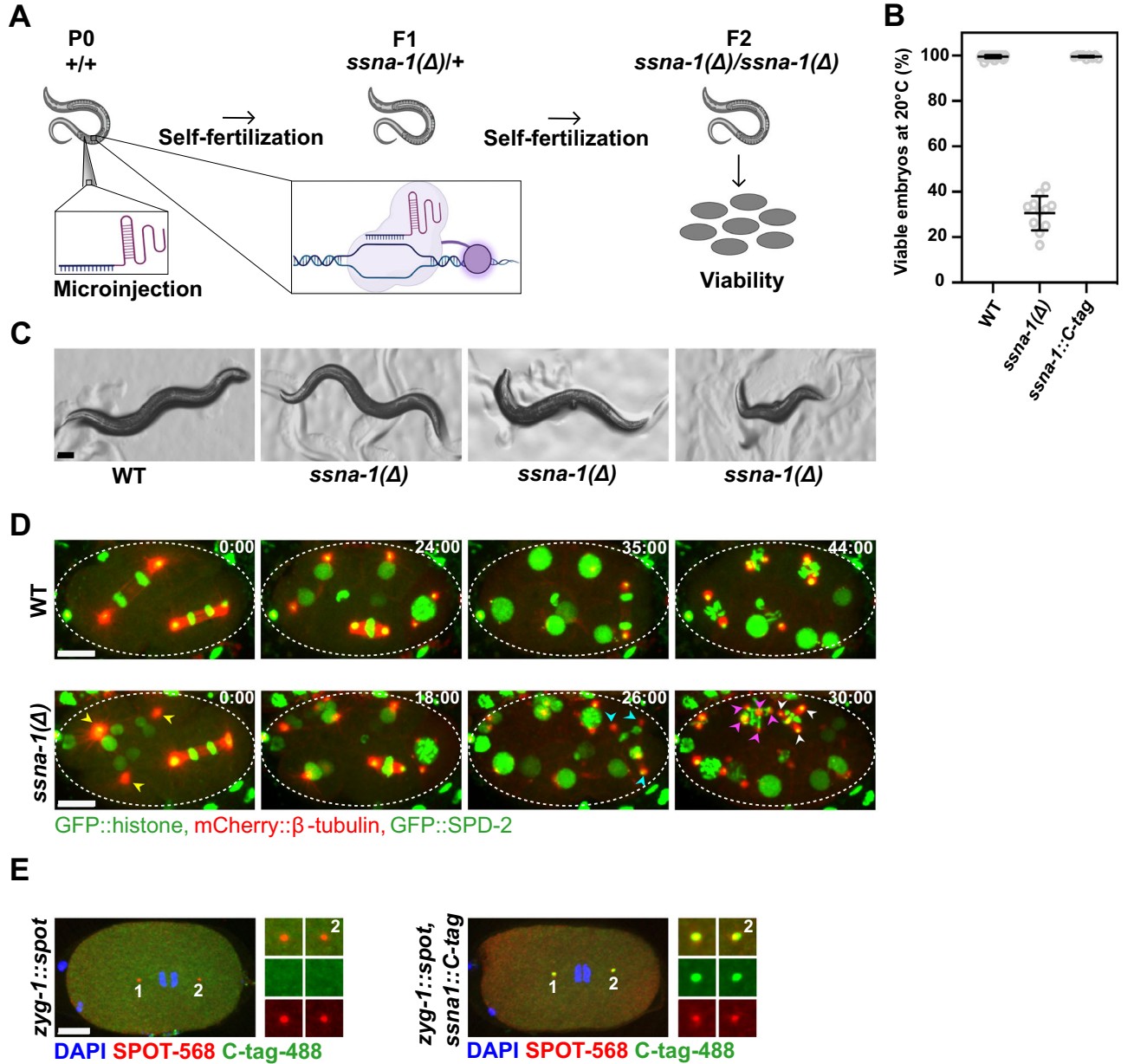

**Fig. 1 | *C. elegans* SSNA-1 is required for embryonic development. A** Schematic of CRISPR/Cas-9-mediated genome editing of *C. elegans* and quantification of embryonic viability. Parts of this figure were created with BioRender.com. https://BioRender.com/7xupr26. This content is not covered by the article's Creative Commons license. **B** Embryonic viability of wild-type, *ssna-1(Δ)*, and ssna-1::C-tag strains at 20˚C. Source data are provided as a Source Data file. *Ssna-1(wt): n* = 29, *ssna-1(Δ)*: *n* = 11, *ssna-1::C-tag*: *n* = 9. Bars depict mean and standard deviation. **C** Representative images of the gross phenotypes of wild-type and *ssna-1(Δ)* strains. Animals homozygous for *ssna-1(Δ)* presented either as wild-type in appearance (left), displayed vulva defects (center) or displayed body length anomalies (right).

Scale bar 100 μm. **D** Representative time-lapse images from 2-cell to 8-cell stages of either wild-type or *ssna-1(Δ)* embryos expressing GFP::histone, mCherry::β-tubulin, and GFP::SPD-2. Arrowheads indicate poles of multipolar spindles that form in *ssna-1(Δ)* embryos, with each color representing a different cell. Scale bars 10 μm. The corresponding movies can be found as Supplementary Movies 1 and 2, respectively. **E** Immunofluorescent images of embryos at early anaphase of the zygote from either *zyg-1::spot* (left) or *zyg-1::spot, ssna-1::C-tag* (right) embryos probed with SPOT and C-tag nanobodies. SSNA-1::C-tag co-localized with the centriolar protein ZYG-1::SPOT. Scale bar 10 μm.

Fig. 4A–D), suggesting that this disordered region corresponds to the N-terminus (residues 1–8) from the two coiled-coil dimers of the eight fibrils in the filament, with the total density accounting for 128 residues (13 kDa). Accordingly, the circular dichroism (CD) spectral analysis indicated that this portion of SSNA-1 is likely disordered, with a higher disordered percentage observed in full-length SSNA-1(1–105) of 27.5%, compared to a truncation construct SSNA-1(12–105) with 9.4% (Supplementary Fig. 4D, S1A, B). Furthermore, these observations were consistent with the AlphaFold prediction of the fragment M1-R18 (Supplementary Fig. 4B) suggesting that residues 1-8 are disordered.

Inter-filament interactions are mediated by the terminal positions of the SSNA-1 between residues 98–105 and 24–28 (Fig. 3B).

The longitudinal connection of the filaments is made by the stacking of two-stranded coiled-coils onto each other (Fig. 3C). Interestingly, the C-terminal helix of each protomer unit forms an overhang and reaches into the adjacent double-stranded helical arrangement creating a triple-stranded junction. At the triple-stranded helical junction connecting neighbouring coiled-coil protomer units, two tandem triple-stranded junctions (junction 1 and 2) were observed (Fig. 3C). The overhanging C-terminus of the first coiled-coil (CC1-a1)

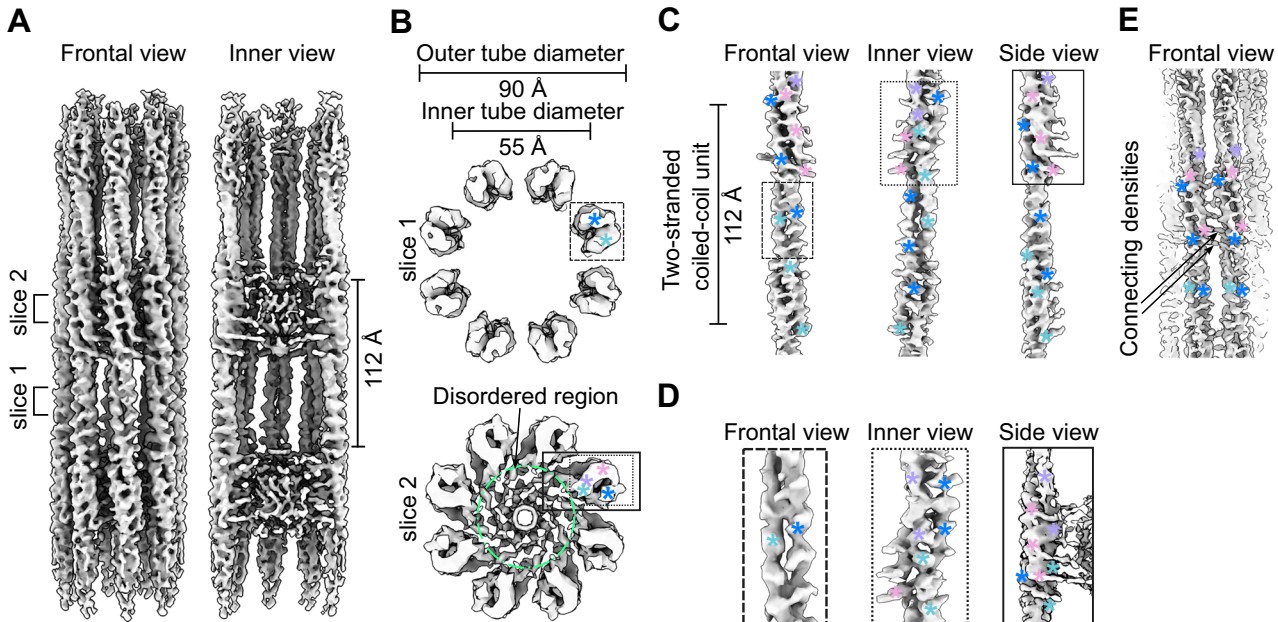

**Fig. 2 | Cryo-EM analysis of SSNA-1(3E). A** Frontal and inner view (longitudinal section) of the SSNA-1(3E) reconstruction. **B** Cross-section view of SSNA-1(3E) filaments at two different slices. Top: Two-stranded coiled-coil region with protomers arranged in 8 fibrils. Bottom: Three-stranded helical junction region with two protomers stacking onto each other. Additional densities extend into the disordered inner lumen (green circle). The boxes indicate regions highlighted in panels (C) and (D). The colored asterisks indicate different helices of protomers. **C** Frontal, inner, and side view of a single SSNA-1(3E) fibril unit (protomer) showing a periodicity of 112 Å. **D** Left: Frontal view of the density representing the core of the protomer (panel B slice 1). Center: inner view of the density representing the connection of two different protomers within the same fibril (three-stranded junction, panel B slice 2). Right: Side view of a three-stranded junction where the density extends to the disordered part (panel B slice 2). The colored asterisks indicate different helices of protomers. **E** Frontal view of densities connecting two different SSNA-1 fibrils.

forms a triple-stranded helix with two alpha helices of the second coiled-coil (CC2) (Fig. 3C). Similarly, the C-terminus of CC2-a2 interacts with the coiled-coil of CC1 (Fig. 3C). The junction interface is formed through a classical coiled-coil structure with predominantly hydrophobic residues in the core and charged residues lined up on the sides (Fig. 3D, E). The key residues in the triple-helical junction are the N-terminal residues Y15, R18E, L19 of one strand and the C-terminal residues L89, Y97, Y105 of the partnering coiled-coil helix. Both parts zip into the corresponding residues of another coiled-coil with the same C-terminal overhang (Fig. 3E). Likewise, the two-stranded coiled-coils are predominantly stabilized by a canonical hydrophobic core (Fig. 3F). Our structural analysis of SSNA-1 self-assembly supports the head-to-tail interaction model proposed for the *C. reinhardtii* homolog of SSNA1[4], where N- and C-termini interact to drive the formation of high-order oligomers. Here, we revealed the register of the coiled-coil and provided a molecular basis for the fibril formation of SSNA1.

The electrostatic surface potential analysis of the 3E mutant shows the presence of negatively charged patches, primarily attributed to the specific 3E mutations (R18E/R20E/Q98E) and the neighbouring charged residues within the three-stranded junctions (Figs. 3C and S4E). While residues R18E and R20E were oriented towards the inner lumen, residue Q98E was located at the interfilamentous interfaces (Supplementary Fig. 4E). These negative charges likely act as a repulsive force to separate the SSNA-1 bundles into 8-stranded filaments, making it amenable to our structural analysis. In the wild-type arrangement these residues can form additional cross-connections between neighbouring filaments to form thick fibers.

### SSNA-1 coiled-coil oligomerization is driven by its termini
To elucidate the critical regions involved in SSNA-1's self-assembly process, we generated various truncations to disrupt fibril formation of SSNA-1 (Fig. 4). The assembly of these variants was assessed using dynamic light scattering (DLS), negative staining EM, and circular dichroism (CD) (Figs. 4, S1A, B). In the case of full-length wild-type SSNA-1(FL-WT, residues 1–105), DLS analysis showed a decay of the autocorrelation function at 1331 μs (Fig. 4C), indicating the presence of a mixed population of small oligomers and filaments with heterogeneous bundles (Fig. 4D).

By analyzing the structural model of SSNA-1(3E) and the residues involved in junction formation (residues 9–22, 89–104) (Fig. 3C), we designed truncation experiments to identify crucial residues involved in self-assembly (Fig. 4A). The truncated SSNA-1(18–105) maintained the ability to form fibrils though they were not bundled and less prominent than the wild-type (Fig. 4A–C, E). In contrast, SSNA-1(19–105), which lacks R18, no longer exhibited fibril assembly (Fig. 4A, F), as shown by DLS (Fig. 4B, F), as well as negative-stain EM (Fig. 4F).

Similarly, we investigated the impact of the C-terminus on the self-assembly process. Considering the interactions between Y15, R18, and Y97 (Fig. 3D), we created truncated fragments, SSNA-1(1–96), SSNA-1(1–97) and SSNA-1(1–100) (Fig. 4A). While SSNA-1(1–100) and (1–97) formed fibrils (Fig. 4A–C, G, H), no fibrils were observed with SSNA-1(1–96) (Fig. 4A–C, I).

Finally, individual constructs with point mutations SSNA-1(R18E/Y97E), (Y15/Y97E), (Y97E), (Y15E), and (R18E), were tested to assess their influence on the oligomerization process. The modifications R18E/Y97E, Y15E/Y97E, and Y97E, disrupted long fibril formation observed for the full-length, and instead thin, shorter fibrils were present (Fig. 4A–C, J–L). The modifications Y15E and R18E lead to precipitated protein and did not allow a comparable assessment of their fibrillar state; however, upon denaturation and refolding, they showed fibril formation (Supplementary Fig. 5B, C). These experiments led us to conclude that the interactions between residues Y97, Y15, and R18 are critical for a stable assembly of triple-stranded junctions and fibril formation. Residues R18 and Y15 complemented each other for

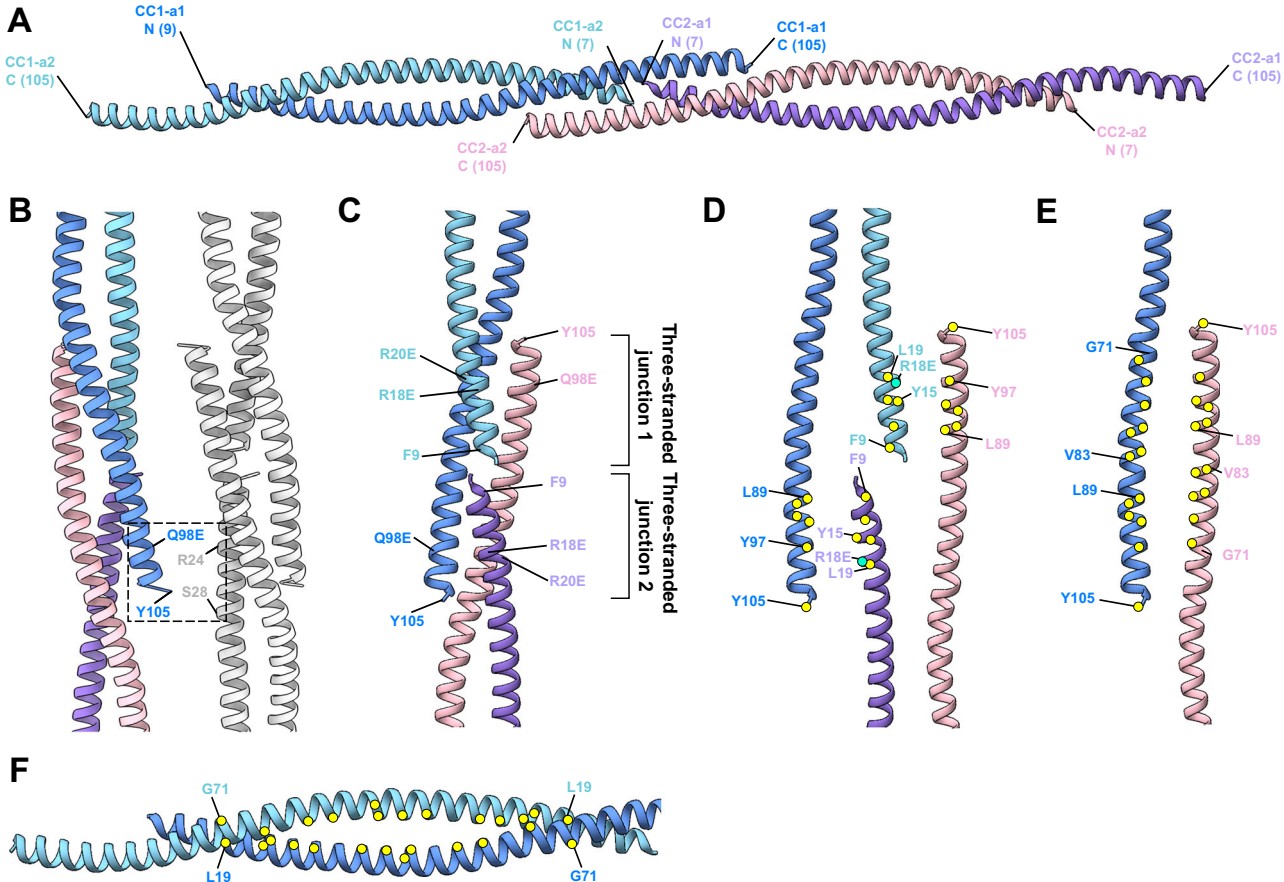

**Fig. 3 | Refined structural model of SSNA-1(3E). A** Ribbon representation of the SSNA-1(3E) structure. Individual fibrils are formed by antiparallel two-stranded coiled-coils that are connected through three-stranded helical junctions. The visible part of the protomers comprises residues 7-105. Individual helices are highlighted in different colors with each N- and C-terminus labeled. **B** Areas connecting individual fibrils near the three-stranded helical junctions. One protofibril is multi-colored and another one is in gray. Additional densities are observed near residues Q98E, Y105 in one fibril and R24, S28 in another. **C** Three-stranded helical junction with two protomers stacking onto each other. Key positions and interactions are highlighted. The point mutations R18E/R20E/Q98E were introduced to obtain the structurally amenable form SSNA-1(3E), forming thin filaments. **D** Exploded view of the two terminal interaction interfaces of the three-stranded helical junctions. Key residues are highlighted with yellow circles for hydrophobic and green circles for charged residues. **E** Exploded view of the third interaction interface of the three-stranded helical junction created by antiparallelly oriented C-termini of two different protomers. Colors are as in (D). **F** The two-stranded coiled-coil regions are stabilized by a canonical hydrophobic core (yellow circles) between residues L19 and G71, respectively.

interaction with Y97 of another fibril. The disruption of fibril formation in the three single-mutants was not as extensive as that observed for the truncated fragments, presumably because of the additional support through hydrophobic interactions at the triple-stranded helical junction (Fig. 3D). Our truncation and mutagenesis analysis of SSNA-1 validates that SSNA-1 self-assembly relies on the triple-stranded helical junctions made of anti-parallel coiled-coil helical overhangs.

## SSNA-1 induces microtubule remodeling through its oligomer formation

The functions of SSNA1 in microtubule remodeling, including microtubule nucleation, elongation and lattice-sharing microtubule branching, have been characterized in previous studies[4,5]. Notably, the observation of microtubule branching is unique to SSNA1, and our previous work indicated that the co-polymerization of SSNA-1 with microtubules provides a guide rail for the splitting of the microtubule lattice and the formation of microtubule branches[4]. Therefore, to investigate the impact of SSNA-1's self-assembly on microtubule binding and branching activity, we performed microtubule binding assays with a subsequent EM analysis (Figs. 5 and S5D–F).

The in vitro co-sedimentation assay of SSNA-1(FL-WT) with pre-polymerized microtubules demonstrated that 70% (14 µM) of SSNA-1 in a 20 µM solution co-sedimented with microtubules (Fig. 5A, B). To assess the region responsible for microtubule recognition and identify whether SSNA-1 self-assembly would be necessary for microtubule binding, we performed binding assays with truncated variants of SSNA-1. The N-terminal truncation SSNA-1(18–105) showed reduced binding to microtubules of 30% (6.0 µM) (Fig. 5A, B). Furthermore, removing one additional residue in SSNA-1(19-105) decreased microtubule binding to only 13% (as 2.6 µM bound) (Fig. 5A, B). These results indicate that R18 and the upstream N-terminal residues of SSNA-1 play a key role in microtubule binding.

To investigate whether the ability of SSNA-1 to bind microtubules depends on fibril formation, we tested the C-terminal truncations SSNA-1(1–96), which cannot form fibrils, and SSNA-1(1–97) or SSNA-1(1–100), which are able to self-assemble (Fig. 4G, H). These C-terminal truncations all bound to microtubules at a similar level, with 41% (8.2 µM), 46% (9.3 µM) and 50% (9.9 µM), respectively (Fig. 5A, B).

Furthermore, to assess whether microtubule-binding would exclusively depend on SSNA-1's N-terminus, we used constructs that abolished the formation of three-stranded junctions (using the mutation Y97E) with additional point mutations at the N-terminus, namely SSNA-1(R18E/Y97E), (Y15E/Y97E), and (Y97E) (Fig. 4J–L). Interestingly, the double modifications SSNA-1(R18E/Y97E) or (Y15E/Y97E), reduced the sedimentation with microtubules to 12% (2.4 µM) and 25% (5.0 µM), respectively, while the SSNA-1(Y97E) single point mutant exhibited

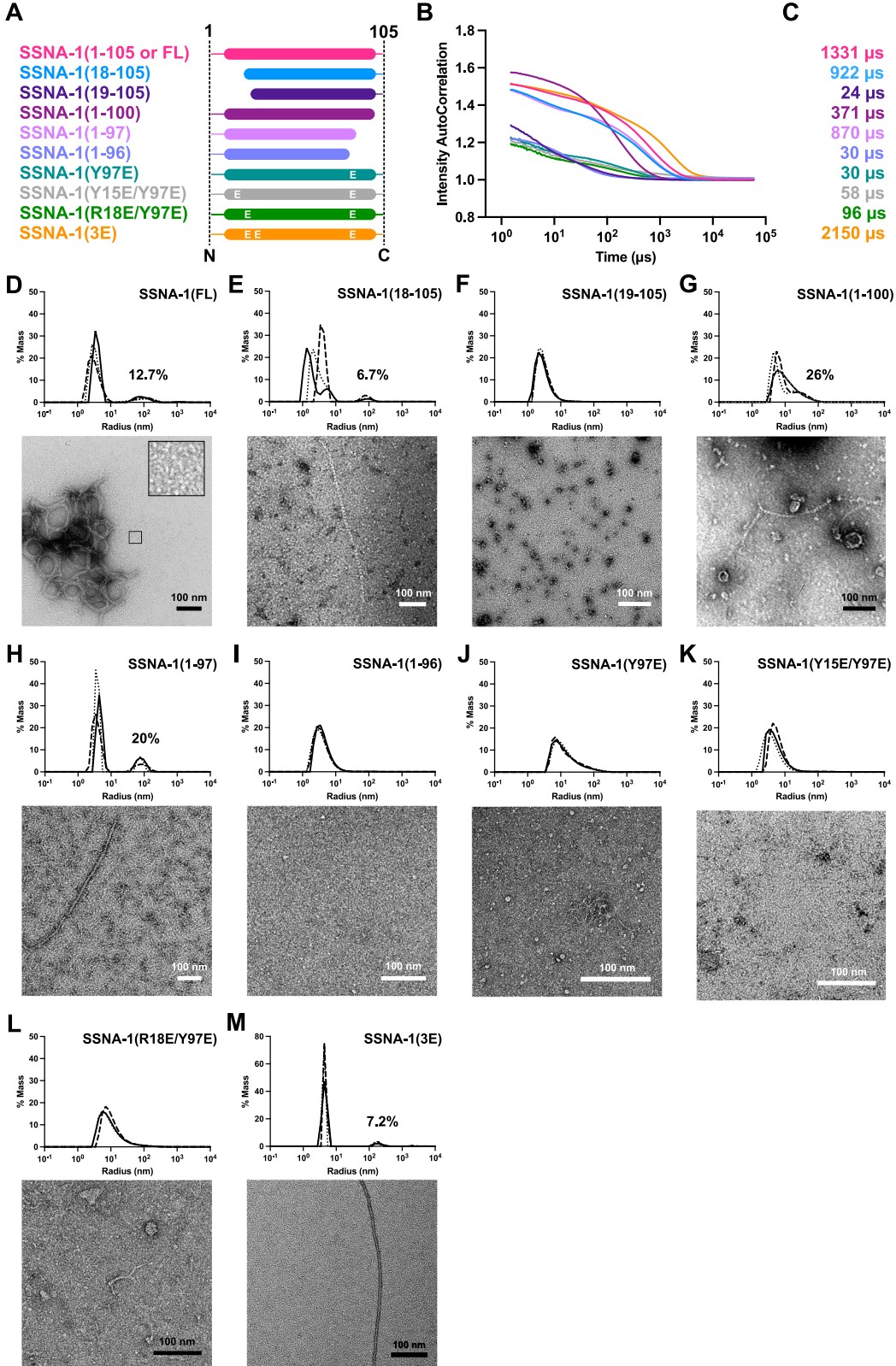

**Fig. 4 | Biochemical characterization of SSNA-1. A** Schematic of SSNA-1 constructs generated to assess functional regions involved in the self-assembly process. **B** DLS autocorrelation curves (cumulative fits) of SSNA-1 constructs and **C** the corresponding decay values of the autocorrelation functions. The decay threshold is considered at an intensity autocorrelation equal to 1. **D–M** Top: Distribution of hydrodynamic radii of SSNA-1 particles calculated from DLS autocorrelation curves represented in B. Data from three individual experiments are plotted. Bottom: Representative negative-staining EM images of the respective SSNA-1 constructs. Scale bars: 100 nm. Source data are provided as a Source Data file.

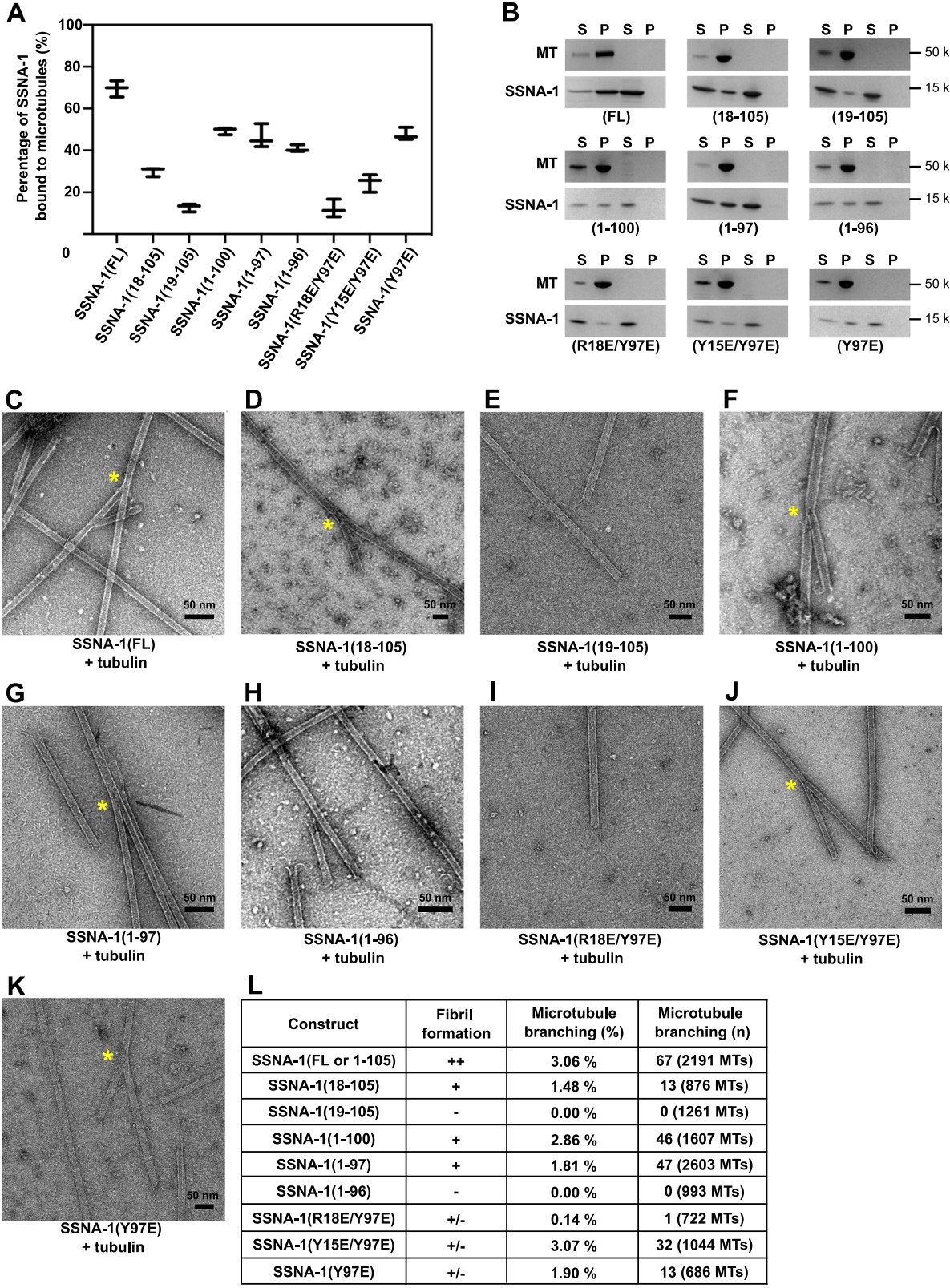

**Fig. 5 | Characterization of the microtubule-binding and -branching activity of SSNA-1 variants. A** Quantification of co-sedimentation assays of pre-polymerized microtubules and different SSNA-1 constructs. The total concentration of each protein was 20 μM (1:1 ratio). The data is represented as whisker plots, with the central line indicating the mean value. Whiskers extend from the minimum to the maximum value. Data points represent results from three independent experiments. **B** Representative SDS-PAGE images of the experiments quantified in panel (A). SSNA-1 controls are on the right, respectively. **C**–**K** Representative negative-staining EM images of different SSNA-1 variants co-polymerized with tubulin at room temperature. The yellow asterisk represents observed microtubule-branching. **L** Summary of the characterization of SSNA-1 variants. Source data are provided as a Source Data file.

48% (9.5 μM) binding (Fig. 5A, B). All these modified proteins cannot form long fibrils as mentioned above. The results demonstrate that the fibril formation is not required for microtubule binding by SSNA-1.

SSNA1 from *C. reinhardtii* has been reported to induce a lattice-sharing microtubule branching when co-polymerized with tubulin[4]. We tested if *C. elegans* SSNA-1 has the same ability, and indeed, co-polymerization of tubulin with wild-type SSNA-1 lead to branched microtubules (Fig. 5C–L). We then demonstrated that the truncated variants SSNA-1(11-105), SSNA-1(18-105), SSNA-1(1–97) and SSNA-1(1–100) that could form fibrils were also able to induce microtubule branching (Supplementary Figs. 4H and 5D, F, G, L). In contrast, fragments SSNA-1(19–105) and (1–96) that were not capable of fibril formation (Fig. 4F, I), were not able to induce microtubule branching (Fig. 5E, H, L). The SSNA-1 (Y97E) mutant, which forms shorter oligomers but not long fibrils (Fig. 4J, K) still retained branching activity, although to a lesser degree (1.9% branched microtubules, compared to 3.1% for the FL-WT, Fig. 5J–L). In contrast, the mutation R18E/Y97E, that formed short fibrils (Fig. 4L) and showed significantly impaired microtubule binding (Fig. 5A, B), abolished microtubule branching (Fig. 5I, L). We conclude that the microtubule branching activity requires the self-assembly ability of SSNA-1 and the binding to microtubules, but it is not necessary for oligomers to make long, bundled fibrils.

### Microtubule recognition and fibril formation are critical for SSNA-1 function in vivo

Our in vitro analysis of SSNA-1 identified key residues for stable fibril formation, particularly the role of Y97 in mediating interactions with residues in neighboring protomers. We also identified R18 as a key residue for microtubule binding. To test the relevance of SSNA-1 self-assembly and microtubule binding in vivo, we introduced a series of point mutations into the endogenous *ssna-1* gene. All three mutations that severely inhibited fibril formation, Y97E, Y15E/Y97E, and R18E/Y97E in vitro lead to embryonic viability comparable to the knockout strain (Fig. 6A, B).

To further examine the importance of the microtubule-binding activity, we analyzed the *ssna-1(R18E)* mutant. While, this mutant exhibited only a mild reduction in embryonic viability to 92% at the permissible temperature of 20 °C (Fig. 6A, B), the viability was significantly reduced to 66% at 25 °C (Fig. 6A, B). Under this condition, recruitment of SSNA-1(R18E) to centrioles was partially impaired, as shown in the ref. 37, indicating that association with centriolar microtubules is essential for the function of SSNA-1. We also characterized the mutant *ssna-1(Y15E)*, which is still able to form fibrils (Supplementary Fig. 5D). This mutant did not result in a significant decrease in viability (99% survival) (Fig. 6A, B). Finally, we tested the viability of the *ssna-1(Y15E/R18E)* double mutant. Given that single mutants of Y15 or R18 retained the ability to interact with Y97 to facilitate filament formation, the double mutant of Y15E/R18E would completely abolish the interaction with Y97, necessary for fibril formation; therefore, we hypothesized that the resulting *C. elegans* mutant would exhibit a severe phenotype. Indeed, the *ssna-1(Y15E/R18E)* double mutant was embryonically lethal, whereas the single mutants ssna-1(Y15E) and *ssna-1(R18E)* exhibited viabilities comparable to WT. Altogether, these findings validate that not only stable microtubule binding, but also stable fibril formation is critical for SSNA-1 function.

## Discussion

The centrosome, a crucial organelle in cell division, relies on coiled-coil proteins to establish scaffolds for the dynamic recruitment of various factors, facilitating centrosome maturation and subsequent cell division[10,27,29,33,38–40]. Here, we solved the structure of fibrillar SSNA-1 and showed that it is a critical member of the centriole and plays a key role in maintaining centriole functionality (Fig. 6C) by following *C. elegans* embryonic development (Fig. 1 and ref. 37).

Our in vitro structural analysis revealed that SSNA-1 forms self-assembled coiled-coil structures by connecting adjacent subunits via C-terminal overhangs that form triple-stranded helical junctions. This junction acts as a hub for fibril formation and microtubule binding, two activities that contribute to SSNA-1's ability to maintain the structural stability of the centriole in *C. elegans* embryos (Fig. 1 and ref. 37). We however note that three point mutations (R18E/R20E/Q98E, 3E) were introduced to facilitate structure determination, as the wild-type C. elegans SSNA-1 formed bundled filaments unsuitable for analysis. Notably, R18 is also crucial for SSNA-1 functionality, including microtubule binding. While this mutation allows structural analysis, it induces physiological disturbance, validated by the *C. elegans* ssna-1(3E) mutant with an average viability of 32% (Fig. 6A), comparable to that of the knockout. Nevertheless, the essential structural mechanism by which the fibril is connected is likely common between the 3E mutant and the wild-type as indicated by the consistency in AlphaFold predictions (Supplementary Fig. 3) and the periodic features observed in the full-length protein (Fig. 4C). Moreover, structure-function analyses, including domain truncations and site-specific mutagenesis, showed defined changes in microtubule binding, validating the functional relevance of key residues. We further verified that removal of the purification tag (His tag) used in this study did not impair microtubule binding (Supplementary Fig. 1C). These findings suggest that while the 3E mutant allows structural elucidation, the filament assembly and higher-order bundling of wild-type SSNA-1 likely play important roles in its full in vivo functionality. While further studies are necessary to elucidate the precise role of the higher order SSNA-1 bundling, we surmise that SSNA-1 may regulate centriolar microtubule stabilization through its self-assembly, resulting in reinforced binding to microtubules during centriole assembly and duplication. Interestingly, our findings also revealed that SSNA-1 localized outside the PCM in satellite structures[37] which are commonly found in vertebrates along with their hallmark scaffolding protein PCM1[37,41,42] but not have been characterized in *C. elegans*. These satellite-like densities diminished with the R18E mutant, which itself showed reduced microtubule-binding. This observation raises the possibility that the bundling ability of SSNA-1 may play a primary role in centriole recognition and stabilization, while its microtubule-binding activity could serve as a secondary function. In the context of SSNA-1's role in *C. elegans* embryonic development and centrosomal regulation, further insights are provided in ref. 37.

Our study furthermore demonstrated the microtubule remodeling activity directed by SSNA-1, wherein microtubule branches split from a single microtubule. The microtubule branching requires SSNA-1 to be fibrillar, which is critical for its in vivo function as well. Our previous report with mammalian neurons expressing microtubule-branching-deficient SSNA1 constructs revealed a significant reduction in axonal branches[4]. Although the physiological significance of SSNA-1's microtubule branching activity remains to be fully elucidated, it is notable that SSNA-1 in *C. elegans* retains branching activity while its sequence homology is rather low compared to the *Chlamydomonas reinhardtii* (25.7%) and mouse (23.8%) proteins[37]. This means, although the protein sequence diverged in *C. elegans*, the SSNA1-induced branching activity is conserved and implies a functional relevance. We speculate that the observed branched microtubules in vitro may represent stochastic intermediate steps in the new production of microtubules under the situation where rapid microtubule polymerization is necessary. During developmental processes requiring rapid cytoskeletal production, the error-prone scaffolding of polymerizing microtubules by SSNA-1 may induce microtubule branching, which may be subsequently repaired as cells mature and transition to a stable operation.

## Methods

### Protein expression and purification

The genes for *C. elegans* SSNA-1 and variants were either synthesized by Twist Bioscience or created by mutagenesis PCR and cloned into

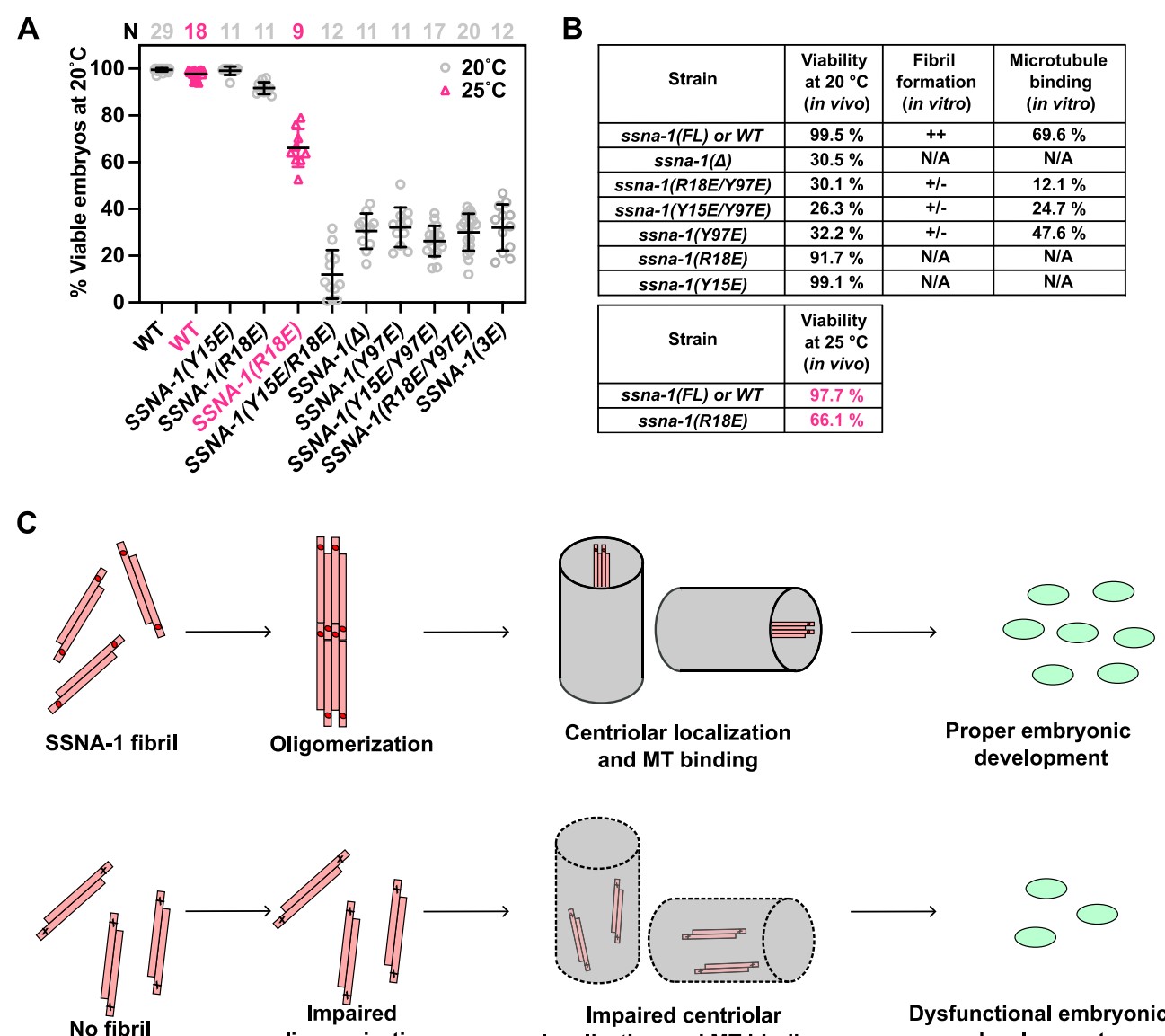

**Fig. 6 | SSNA-1's ability to form fibrils dictates *C. elegans* viability during its embryonic development. A** Embryonic viability of strains homozygous for different *ssna-1* alleles. Each data point shows the percentage of viable embryos from a single hermaphrodite. Bars depict mean and standard deviation. The number (N) of counted hermaphrodites for each allele is given on the top of the graph. **B** Comparison of the viability of *C. elegans* mutants in vivo with the characterization of different SSNA-1 constructs in vitro. Fibril formation is assessed as in Fig. 5L. Microtubule-binding ability was taken from Fig. 5A, B. **C** Model for the molecular function of SSNA-1 in stabilizing centrioles in *C. elegans*. Top: SSNA-1 stabilizes centrioles only when it is able to form filaments and bind to microtubules. Filaments (coiled-coils) are depicted as pink lines with red dots indicating the position of key residue R18. Bottom: When residue R18 is mutated (indicated by a black x), fibril- and oligomer formation is impaired, and worms show low viability as centrioles lose their integrity. The location of SSNA-1 within centrioles is based on the expansion microscopy results shown in ref. 37. Details of in vivo function of SSNA-1 are available in ref. 37. Source data are provided as a Source Data file.

homemade bacterial expression vectors pCB-a-bax by ligation-independent cloning or Gibson assembly. All SSNA-1 constructs were prepared as fusion proteins containing 6x-His-tag with a HRV 3Ccleavage site (MGKHHHHHHGSLEVLFQGP) at the N-terminus. SSNA-1 proteins were expressed in *Escherichia coli* BL21(DE3) (Agilent Technologies Inc.) using TB media and induction with 0.4 mM IPTG overnight at 18 °C. Purifications were performed as based on the report[4]. Cells were harvested and sonicated with a Branson SFX250 Sonifier with 102-C Converter and a 3/4" High-Gain Horn in cold lysis buffer containing 50 mM sodium phosphate-NaOH pH 7.5 (Oakwood #099682, VWR #97064-480), 150 mM NaCl (Millipore #1.06404), 10% (v/v) glycerol (Sigma #G7757), 1 mM DTT (AppliChem #A1101) supplemented with protease inhibitors 1 mM pepstin A (AppliChem#A2205), 1 mM PMSF (BioVision #1548-5) and 1 mM leupeptin

(AppliChem #A2183) or cOmplete EDTA-free Protease Inhibitor Cocktail (Sigma #5056489001) and 10 mM imidazole (Thermo Fisher Scientific #O3196500). After clarification in a JA-25.50 rotor (Beckman #363055) at 25k rpm, proteins were purified by His-affinity chromatography (cOmplete His-Tag Purification resin, Sigma #5893682001) from the supernatant. Proteins were eluted with 50 mM sodium phosphate-NaOH pH 7.5, 150 mM NaCl, 10% glycerol (v/v), 1 mM DTT, 500 mM imidazole, and then dialyzed against lysis buffer at 4 °C. After dialysis, samples were centrifuged at 16,100$g$ and 4 °C for 10 min to remove large oligomers. The protein concentrations were determined using the extinction coefficients at 280 nm (https://web.expasy.org/protparam/). Proteins that required additional purification steps were subjected to ion exchange chromatography using HiTrap Q HP (Cytiva #17115401) or HiTrap SP HP (Cytiva #17115201) columns and linear NaCl

gradients. Non-filamentous variants SSNA-1(19–105) and SSNA-1(1–96) were further purified by size-exclusion chromatography on a Superdex 200 increase 3.2/300 column (Cytiva #28990946). The masses of all proteins were confirmed by LC/MS measurements. When required, samples were concentrated to 4–5 mg/ml using Amicon ultrafiltration devices with 10 kDa MWCO (Millipore #UFC501096), flash-frozen in liquid nitrogen, and stored at −80 °C. The constructs, SSNA-1(R15E) and SSNA-1(R18E) were insoluble and hence purified as previously described supplementing lysis and elution buffer with 8 M urea (MP Biomedicals #210569505). The proteins were refolded by dialysis against lysis buffer. Unfolded and large oligomers were removed by centrifugation at 16,100$g$ and 4 °C in an Eppendorf 5417R centrifuge for 10 min. Tubulin was purified from porcine brains (a generous gift of the Bayerische Landesanstalt für Landwirtschaft, Garching) following a previously published protocol[43].

### Circular dichroism
Circular dichroism (CD) spectroscopy of SSNA-1 constructs was performed on a Jasco J-715 equipped with a Peltier thermostat. Measurements were obtained in cuvettes (High Precision Cell Quartz Suprasil) of 0.1 cm optical path length over the range of 195–250 nm at 4 °C, with 100 mdeg of sensitivity, continuous scanning mode, 100 nm/min scanning speed, and 4 spectra accumulation. Secondary structure predictions were estimated with CDSSTR[44]. Unfolding curves were obtained by following the change in the CD intensity signal at 222 nm as a function of temperature over the range of 4–90 °C with a heating rate of 60 °C/h. Proteins were measured at a final concentration of 0.1 mg/mL.

### Dynamic light scattering (DLS)
Measurements were performed on a Wyatt DynaPro NanoStar DLS (Wyatt) at a wavelength of 658 nm and a scattering angle of 90°. All measurements were done at a final protein concentration of 0.1 mg/ml in triplicates with 10 acquisitions each for 1 min (acquisition time 5 s and read interval 1 s) at 20 °C. Cumulant and regularization fits were processed with DYNAMICS (Wyatt) and plotted with PRISM 10 (GraphPad).

### Sedimentation assay
Microtubule-binding of SSNA-1 was tested by sedimentation of purified proteins. SSNA-1 constructs were pre-sedimented twice at 16,100$g$ and 25 °C using a Beckman TLA-120.2 rotor to remove large oligomers for 10 min and 5 min, respectively. Porcine brain tubulin was pre-polymerized for 30 min at 37 °C and a final concentration of 60 µM in BRB80 buffer containing 80 mM PIPES/KOH pH 6.8 (Sigma #P6757, VWR #BDH9262), 1 mM MgCl$_2$ (Millipore #1.05833), 1 mM EGTA (Sigma #E3889) supplemented with 1 mM GMPCPP (Jena Bioscience #NU-405). SSNA-1 was dialyzed against BRB10 buffer (PIPES/KOH 10 mM pH 6.8, 1 mM MgCl$_2$ and 1 mM EGTA) at 4 °C for 5 h, and incubated with pre-polymerized microtubules (MTs) for 4 min at room temperature. The final concentration for each protein was of 20 µM. After incubation, the reactions were sedimented at 9777$g$ and 25 °C in a Beckman TLA-120.2 rotor for 10 min. Supernatants and pellets were analyzed by SDS-PAGE and imaged with a Bio-Rad GelDocGo imager. Band intensities were measured and analyzed in FIJI[45], EXCEL (Microsoft) and PRISM 10.

### Microtubule branching analysis
30 µM SSNA-1 fragments were co-polymerized with 15 µM purified brain tubulin in BRB80 buffer supplemented with 1 mM of GMPCPP for 5 min at RT. and the mixture was observed under negative stain EM. The specimens were incubated on carbon-coated grids for 1 min at RT, washed twice with BRB80 buffer and stained with uranyl acetate or formate for 1 min. Images were acquired using a FEI Tecnai T12 electron microscope at 120 kV at different magnifications.

### Strains and maintenance of *C. elegans*
Worms were maintained on *E. coli* OP50 seeded MYOB agar plates according to standard protocols[46] All strains used in this study are listed in Supplementary Table 2.

### CRISPR-Cas9 genome editing
CRISPR-Cas9 mediated genome editing was performed as previously described[47,48]. Oligonucleotide repair templates and primers were purchased from Integrated DNA Technologies, and crRNAs were purchased from Dharmacon. Cas9 protein was purchased from PNA Bio (#CP01). All repair templates and crRNA sequences used in this study are listed in Supplementary Table 3.

### Embryonic viability assay
For embryonic viability, L4 larvae were isolated to MYOB plates and incubated at 20 °C for 24 h. The gravid adults were then singled onto 35 mm plates and allowed to lay eggs for 24 h. Adults were then removed, and embryos were allowed another 24 h to hatch at 20 °C, after which living larvae and dead eggs were quantified.

### Fixed and live imaging
Immunofluorescence and time-lapse imaging of embryos was performed as previously described[37,48]. The C-tag was detected using CaptureSelect Alexa fluor 488 anti-C-Tag (Thermo Fisher Scientific #7213252100) and the SPOT tag was detected using SPOT-Label Alexa Fluor 568 (Proteintech #ebAF568). Both VHH nanobodies were used at a 1:1,000 dilution and incubated for 2 h at room temperature. Both fixed and live images were obtained using a Nikon Eclipse Ti spinning disk confocal microscope using a Plan Apo VC 60×1.4 NA oil immersion lens, a CSU-X confocal scanning unit (Yokogawa Electric Corporation), and an Orca-FusionBT C15440 digital camera (Hamamatsu Photonics). 405 nm, 488 nm, and 561 nm solid state lasers were controlled via a Nikon LunF-XL laser module (Nikon Instruments,) and images were acquired using the NIS-elements software (Nikon Instruments). Images were processed using FIJI.

### Negative-staining electron microscopy
Homemade carbon-coated grids were prepared, and glow discharged before use in a PELCO easiGlow (Ted Pella) for 45 s with a plasma current of 15 mA. SSNA-1 fibrils were grown through overnight dialysis against lysis buffer at 4 °C (see above). 5 µl SSNA-1 proteins at a final concentration of 0.01 mg/mL were applied onto grids, washed with H$_2$O and stained with 1% (w/v) uranyl acetate (Electron Microscopy Sciences #22400) or uranyl formate (Electron Microscopy Sciences #22450). Dilutions were performed with lysis buffer when required.

### Cryo-electron microscopy and image processing
SSNA-1(3E) was dialyzed overnight at 4 °C against lysis buffer (see above) to grow filaments. 5 µl sample at a final concentration of 0.1 mg/mL was applied to glow discharged Quantifoil grids R2/1. After 1 min incubation at room temperature, the grid was manually blotted on the side of the grid with filter paper (Whatman Grade 1, Cytiva #1001-055) and blotted again with 4 µL of a solution containing lysis buffer supplemented with 0.1%(v/v) n-octyl-β-D-glucopyranoside (β-OG, Anatrace #O311). The grid was then transferred to an EM GP2 plunger (Leica) with a chamber temperature of 15 °C, maximal humidity, incubated for 1 min, blotted with Grade 595 filter paper (Ted Pella #46000-200) for 6 s and vitrified in liquid ethane. Data was collected on a Krios G4 transmission electron microscope (Thermo Fisher Scientific), equipped with a Bio-Continuum energy filter (Gatan, slit width 20 eV) and K3 Summit direct detector (Gatan) at an acceleration voltage of 300 kV. Movies were recorded using EPU (Thermo Fisher Scientific) in super-resolution mode at a magnification of 105 kx corresponding to a raw pixel size of 0.412 Å/pixel. The defocus range varied from −0.5 to −2.3 µm with 40 frames per movie and a total dose of 40.84 e$^-$/Å$^2$. In total, 10,396 movies were used

for further analysis in CRYOSPARC[49]. Movies were motion corrected (patch motion correction), adjusted with a Fourier cropping factor of 0.5 resulting in a pixel size of 0.824 Å and CTF measurements were performed via patch CTF. 8390 exposures were selected for further processing. An initial set of 888,099 helical particles were auto-picked with filament tracer, extracted with a box size of 512 pixel, F-cropped to a box size of 256 pixel (resulting in a pixel size of 1.648 Å), and sorted via 2D classification. Ab initio reconstructions and helical refinements without supplying helical parameters were performed using C1 as point group symmetry to obtain the initial helical model. The obtained particles were subjected to multiple rounds of 3D classification and heterogenous refinement (Cryo-EM workflow in Supplementary Fig. 2). 84,939 particles were selected for local CTF refinement and further rounds of 3D classification and heterogenous refinement to yield 80,664 particles, which were used for helical refinement using C8 as point group symmetry. Our analysis required extensive classification due to the intrinsic heterogeneity of the fibrils. Finally we evaluated whether the particles used for the final map have chirality issue. We performed two independent ab initio refinements and validated that the Euler angle distributions converged similar to each other, indicating that the structure is determined with the correct chirality. Applying symmetry expansion followed by 3D classification, and local refinement with C1 symmetry did not yield an improvement of the resolution, also indicating the heterogenic nature of the fibril.

To determine the length asymmetric unit, particles were re-extracted with a box size of 1024 pixel, F-cropped to 512 pixel (resulting pixel size of 1.648 Å/pixel), and an averaged 2D class was used to calculate a power spectrum using FIJI[45] (Supplementary Fig. 2C). All 3D reconstructions were visualized in CHIMERAX[50].

### Hybrid model building
AlphaFold 2.3.2[51] was used to predict a SSNA-1(3E) model through ColabFold[52]. The resulting model was then fit to the cryo-EM map in CHIMERA, and further rigid-body fit in COOT[53] and CryoFit in PHENIX[54]. The resulting model showed a clash of residues M1-S8 that were predicted to be part of the coiled-coil by AlphaFold (Supplementary Fig. 3A). We observed an extra density for residue F9 in our cryo-EM map protruding inward the filament, connecting to the disordered region in the inner lumen (Supplementary Figs. 3A and S3B). Based on the available information we prepare a starting model for SSNA-1(3E) comprising residues 7–105 and performed real-space refinement and validation in PHENIX (Supplementary Table 1). Electrostatic surface potential maps were generated with the Adaptive Poisson-Boltzmann Solver (APBS)[55].

### Reporting summary
Further information on research design is available in the Nature Portfolio Reporting Summary linked to this article.

## Data availability
The 3D reconstruction of SSNA-1(3E) and the structure coordinates generated in this study have been deposited in the EMDB and PDB database under accession codes EMD-47147 and PDB 9DSM. Source data are provided with this paper.

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

## Acknowledgements

We thank the members of the Mizuno lab for many helpful discussions. We acknowledge the NIH Multi-Institute Cryo-EM Facility (MICEF), the NHLBI Biophysics Core, and the NHLBI Biochemistry Core for infrastructure. This work used the computational resources of the NIH HPC Biowulf cluster (http://hpc.nih.gov). This work was supported by the Division of Intramural Research Program of the National Heart, Lung, and Blood Institute (1ZIAHL006264) and the National Institute of Arthritis and Musculoskeletal and Skin Diseases of the National Institutes of Health. Research within the KOC lab was supported by the Intramural Research Program of the NIH, The National Institute of Diabetes and Digestive and Kidney Diseases (NIDDK). The contributions of the NIH author(s) were made as part of their official duties as NIH federal employees, are in compliance with agency policy requirements, and are considered Works of the United States Government. However, the findings and conclusions presented in this paper are those of the author(s) and do not necessarily reflect the views of the NIH or the U.S. Department of Health and Human Services.

## Author contributions

L.A., J.A.P., C.B. and N.B. performed experiments and analyzed the data. L.A., J.D., R. Z., C.B. and N.M. performed computational analysis. C.B., K.F.O. and N.M. supervised the work.

## Funding

## Competing interests

The authors declare no competing interests.
