## [Transparent Peer Review file · Nature Communications]

Structural insights into SSNA1 self-assembly and its microtubule binding for centriole maintenance

Corresponding Author: Dr Naoko Mizuno

Version 0:

Reviewer comments:

Reviewer #1

(Remarks to the Author)

Summary:

This manuscript by L. Agostini et al. represents a very beautiful and comprehensive study combining cryo-EM, mutagenesis, and in vivo functional studies to elucidate both the structure and functional roles of SSNA-1 in *C. elegans* in cells. Benefiting from a smart design of the mutant SSNA-1, the SSNA-1(3E) with R18E/R20E/Q98E, The authors revealed the self-assembled coiled-coil fibrillar structure of SSNA-1 and its role in microtubule binding. They identified the key interaction sites critical for the filament formation and performed in vivo functional studies to show how these key residues affect centrosome maintenance and embryonic viability. The combination of structural biology, biochemical assays, and functional validation makes this work truly impressive, providing significant insights into the molecular mechanisms of SSNA-1. I have no major concerns about the study and quality of the results overall and I highly recommend publication of this work by addressing a few minor issues as below.

1. Differences between wildtype and mutant structures:

The authors introduced the R18E/R20E/Q98E mutations in SSNA-1, termed SSNA-1(3E), to achieve a more technically feasible form for cryo-EM analysis. While similar approaches have been widely applied in different biological systems, I am concerned that these mutations may induce unexpected structural artifacts that largely deviate from the wild-type assembly. I fully understand this is a major technical consideration in structural biology as many people do similar things as well, but I found there are limited discussions about this issue. I raised this question because the higher-order assemblies of the WT and mutants seem to be quite different. I would recommend the authors explicitly discuss the impacts of the mutations on fibril stability, flexibility, and physiological relevance etc. Also, I was wondering if there is any in situ evidence to show the existence of the reported fibril formation using WT SSNA-1. This is not a requirement for this current study.

2. Resolution issues due to imposed C8 symmetry:

The authors refined their cryo-EM structure using the C8 symmetry, which resulted in a resolution of approximately 4.5 Å. I am concerned that imposing C8 symmetry might introduce a bias, particularly if the true filament structure deviates from a perfect C8 symmetry. This may explain why the resolution was not even better, despite the final 640,000 particles (8x8K). This resolution is quite tricky in terms of atomic model building. I was wondering if the authors could try to make a bit more efforts to improve the resolution. For example, I realized that the authors used C8 to refine the final map and stopped there. I felt that based on the C8-symmetrized map, a symmetry expansion, followed by focused refinement and classification using C1, and re-refinement of the best classes might help. Also, just out of intuition, I felt that most cryo-EM programs dealing with a structure like this filament would confuse the chirality of the structure, leading to an average of the correct and Z-flipped maps. This is just my guess. There is a very simple way to evaluate if this was the case: perform two independent ab initial refinements (ideally from the very beginning, which avoid inter-dependence of the references used) and compare the final Euler angle distributions of individual particles from the two independent refinements. In the ideal case, every single particle will converge to the same Euler angles, and in the worst case, particles are just randomly classified into the correct and false hands.

Other comments:

Line 26: "the underlying molecular mechanism have remained unclear..." → should be singular form, "mechanism has remained unclear."

Line 35: "...significant reduction in embryonic viability and the formation of multipolar spindles..." No major issue and I did understand what the authors intended to say here, but I felt the sentence can be a bit confusing for general readership because I could also interpret this as "reduction in both A (embryonic viability) and B (the formation of multipolar spindles)". Something like "reduction in embryonic viability, along with the formation of multipolar spindles" may resolve this confusion.
Line 214: "assembly of these variants were assessed using dynamic light scattering (DLS)", should be 'was assessed'.
Line 367: "Accession Codes" → It seems "Data Availability" is the standard term in Nature press.

Reviewer #2

(Remarks to the Author)

This study by Agostini et al. dissected the structure of microtubule-associated protein SSNA1 through cryo-EM, uncovering key residues (or regions) for effective self-assembly of SSNA1 fibril, microtubule binding and remodeling, and centriole functionality. Initially, they generated *ssna-1* knockout nematodes and observed significant embryonic lethality, underscoring its important physiological functionality. They purified *C. elegans* SSNA-1 (3E) mutants for cryo-EM-based structure determination, and found SSNA-1 forms double-stranded coiled-coils where intra-filament junctions were triple-stranded. Through analysis of truncated SSNA1 mutants, they identified several key residues such as Y15, R18 and Y97, which are crucial for long fibril formation and microtubule binding. Remarkably, these residues (or SSNA1 fibril formation) are also indispensable for in vitro microtubule branching induced by SSNA1. Finally, this study demonstrated that fibril formation and microtubule binding activity of SSNA-1 are imperative for its physiological roles including centriole maintenance.

These results offer significant advances in our understanding of the working mechanism of SSNA1 in centriole functionality and microtubule branching by providing structural and molecular insights. However, there are some critical issues remaining to be addressed before publication.

Major comments:

1) The authors identified an SSNA-1 mutant with 3 point mutations (R18E/R20E/Q98E or 3E) for further structure determination, as wild-type *C. elegans* SSNA-1 formed bundled filaments that were not suitable for analysis. However, they also found that several residues, including R18, are crucial for the functionality of SSNA1 including microtubule binding. This indicates that SSNA-1 (3E) may not fully represent the physiological state of SSNA1 in vivo. Does *ssna-1* (3E) mutant animals exhibit significant embryo lethality as *ssna-1* KO animals? At least, how the physiological functionality of SSNA-1 (3E) mutant diverge from wild-type SSNA-1 should be discussed.

2) Line 233. "The modifications R18E/Y97E, Y15E/Y97E, and Y97E, disrupted long fibril formation observed for the full-length, and instead thin, shorter fibrils were observed (Fig. 4A-B, 4I-K)." However, in Figure 4B and 4J, it appears that Y15E/Y97E still forms long fibrils as wild-type SSNA1.

3) Previously, the authors' group identified that SSNA1 directly induced microtubule branching in vitro, although a subsequent study by another group failed to observe SSNA1-induced MT branching (Lawrence et al., 2021, eLife). In this study, the authors illustrated that 4.29% (7 of 163) of MTs were branched induced by FL SSNA-1. Furthermore, SSNA-1 mutants those failed to form long fibrils show deficiency in inducing MT branching. Considering the relatively low branching frequency, I recommend the authors analyze a larger number of MTs in each group for statistics (e.g., 500-1000 MTs per group). This would make their conclusion more authentic.

Minor comments:

1) Line 31: "..., highlighting that self-assembly of SSNA-1 facilitates effective microtubule interaction by creating hubs along a fibril."

This conclusion is not directly supported by their results.

2) MT branching assay is missing in Methods section.

3) Line 147: "we purified *C. elegans* SSNA-1 and...". It would be better to specify that SSNA-1 was purified from bacteria.

4) Line 329: "...it is a critical member of the centriole coiled coil fibrillar protein with its key role in maintaining centriole stability". However, no evidence in this study shows that SSNA-1 is critical for centriole stability. Would it be more accurate to replace 'stability' with 'functionality'?

5) Line 439: "for 5' " Does this mean "for 5 min"?

6) The authors deposited too many conclusions into figure legends. For example:

Line 616: "The results indicate that the formation of fibrils is not essential for the binding of SSNA-1 to microtubules, however, the SSNA-1 N-terminus, including residue R18 and its neighbouring residues, is critical for microtubule-binding."

Line 628: "These results indicate that microtubule branching requires SSNA-1 to bind microtubules and form oligomeric or fibrillar assemblies."

Line 638: "These results show that residues Y15, R18, Y97, which are important for fibril formation, are also critical for the

viability of *C. elegans*.”

These conclusions should be removed from figure legends.

Reviewer #3

(Remarks to the Author)

In this manuscript Agostini et al. investigate the self-assembly properties of *C. Elegans* SSNA-1 protein that associates with microtubules and is implicated in the structural stability of centrosomes (accompanying paper). Using mutated SSNA-1 constructs, the authors report that mutant SSNA-1 forms anti-parallel coiled coils which then assemble into higher-order fibrils forming specific triple-stranded helical junctions. The authors find that these triple-stranded junctions are also sites of microtubule interaction. Furthermore, the authors investigate the ability of several mutant and deletion SSNA-1 constructs to form microtubule branches, previously reported for wt *C. reinhardtii* SSNA-1 homolog. Overall, the manuscript provides little substantive insight into the mechanism by which SSNA-1 supports centriole structure – a function reported in the accompanying paper. As such, the limited results of the present manuscript could be fully absorbed into the other SSNA-1 paper by the same group of authors.

Specific comments:

Figure 1: New results in this paper originate from the subsequent structural studies; as such, current Figure 1 does not fit into the present manuscript. Rather, these results are covered in the accompanying manuscript, where they seem to be more appropriate.

Figure 2. Given the significant mutations in the protein that were induced in order to determine the fibril structure, it is unclear how physiologically relevant is this 8-filament tube structure? Notably, the exact residues that were mutated here are implicated in fibril formation. Thus the reported structure may well be an artifact of the induced mutations. Furthermore, how/where would this fit on the microtubule and within the centriole? Note the apparent discrepancy from the previously reported structure by Basnet et al. – this should be addressed.

Figure 3. This is a fully alphafold figure which should be incorporated within or as a supplemental to Figure 2. Also, the structure appears contradictory to previously published structure by the authors (reporting parallel, as opposed to antiparallel dimers). Furthermore, the authors note that R18 is a key residue for the triple helical junction but they mutated it for their structural determination. If this is a key residue then it would be helpful to have a structure that actually reflects this. Can the authors predict what the structure would look like without induced mutations?

Figure 4. Analysis of the 3E mutant should be included here. Also, the subsequent categorization of fibril formation should be done based on these DLC results. Can the spectra be quantified?

Figure 5. This is a particularly weak figure as it reports extremely rare occurrences of what the authors report as microtubule branching and correlate this with the potential for fibril formation. The statistics are very poor, the images of branching tenuous at best, and the finding that these events are so rare puts in question the physiological relevance of these observations.

Figure 6. This figure fails to provide a cohesive picture of the results. It is completely unclear how the structures presented in Figures 2 & 3 fit into this schematic? The binding of SSNA1 fibrils to microtubules as well as within centrioles has not been clarified by the presented data.

Version 1:

Reviewer comments:

Reviewer #1

(Remarks to the Author)

The revised manuscript is excellent. I have no more questions.

Reviewer #2

(Remarks to the Author)

This study by Agostini et al. dissected the structure of microtubule-associated protein SSNA1 through cryo-EM, uncovering key residues (or regions) for effective self-assembly of SSNA1 fibril, microtubule binding and remodeling, and centriole functionality.

The authors have well addressed the problems that I raised before. I have no further concerns about this study.

Reviewer #3

(Remarks to the Author)

In their revised manuscript, the authors have adequately addressed my originally identified concerns.

We sincerely thank all the reviewers for their constructive criticisms and suggestions. Below are the point-by-point responses of their comments.

Reviewer #1 (Remarks to the Author):

Summary:

This manuscript by L. Agostini et al. represents a very beautiful and comprehensive study combining cryo-EM, mutagenesis, and in vivo functional studies to elucidate both the structure and functional roles of SSNA-1 in C. elegans in cells. Benefiting from a smart design of the mutant SSNA-1, the SSNA-1(3E) with R18E/R20E/Q98E, The authors revealed the self-assembled coiled-coil fibrillar structure of SSNA-1 and its role in microtubule binding. They identified the key interaction sites critical for the filament formation and performed in vivo functional studies to show how these key residues affect centrosome maintenance and embryonic viability. The combination of structural biology, biochemical assays, and functional validation makes this work truly impressive, providing significant insights into the molecular mechanisms of SSNA-1. I have no major concerns about the study and quality of the results overall and I highly recommend publication of this work by addressing a few minor issues as below.

We thank the reviewer 1 for recognizing our efforts to comprehensively characterize the SSNA1 function mechanistically as well as biologically.

1. Differences between wildtype and mutant structures:

The authors introduced the R18E/R20E/Q98E mutations in SSNA-1, termed SSNA-1(3E), to achieve a more technically feasible form for cryo-EM analysis. While similar approaches have been widely applied in different biological systems, I am concerned that these mutations may induce unexpected structural artifacts that largely deviate from the wild-type assembly. I fully understand this is a major technical consideration in structural biology as many people do similar things as well, but I found there are limited discussions about this issue. I raised this question because the higher-order assemblies of the WT and mutants seem to be quite different. I would recommend the authors explicitly discuss the impacts of the mutations on fibril stability, flexibility, and physiological relevance etc. Also, I was wondering if there is any in situ evidence to show the existence of the reported fibril formation using WT SSNA-1. This is not a requirement for this current study.

Reviewer raised an excellent point of discussing potential differences between the 3E mutant and WT in structural and functional aspects. We agree with the reviewer that the R18E mutation and the lack of bundling, while feasible for structural analysis, is likely to cause physiological disturbances. To address this point, we have created the c.elegans SSNA-1 (3E) mutant and quantified its viability. The 3E mutant has an average viability of 32%, similar to the knockout (now added to Fig. 6A). We nevertheless think the essential structural mechanism of how the fibril is connected is retained, as consistently shown with

alpha-fold prediction (now added to Fig. S3) as well as the periodical features that are displayed in the FL (Fig. 4C). The reviewer is correct that we have a limited understanding of the ability of WT SSNA-1 to make a thicker bundle and the relevance of it. We also think that, based on the enigmatic localizations of SSNA1 at cilia, basal body, and at the centrosome satellite, there is a possibility that SSNA1 may have multiple localizations with overlapping but distinct functions and interaction partners. Curious observation of c.elegans R18E mutant that loses the localization at the satellite might indicate that the bundling of SSNA1 is important for the recognition of centriole and its stabilization but the microtubule binding may be a secondary function for centriole recognition.

Together with the new added results (Fig. 6 and Fig. S3), the discussion about the limitation of our structural insights and the necessity of understanding the functional relevance of the SSNA-1 bundling in the future is now added in discussion.

Regarding the in situ direct observation of SSNA1 using cryo-EM or cryo-ET, we have not yet identified SSNA1 within cells. While this remains an aspiration for us, we have not been able to visualize it directly. That said, we have noted the presence of in situ macromolecular organizations resembling the in vitro SSNA1 structures (bundled filaments and thin filaments). However, robust identification requires advanced labeling techniques, which remain an active area of research in the cellular cryo-ET field. The current state of technology does not yet allow for routine analysis of this nature. Overall, annotating in situ structures demands careful assessment, including labeling and structural analysis, to ensure accurate identification. This sort of analysis will be a key focus of our future research.

2. Resolution issues due to imposed C8 symmetry: The authors refined their cryo-EM structure using the C8 symmetry, which resulted in a resolution of approximately 4.5 Å. I am concerned that imposing C8 symmetry might introduce a bias, particularly if the true filament structure deviates from a perfect C8 symmetry. This may explain why the resolution was not even better, despite the final 640,000 particles (8x8K). This resolution is quite tricky in terms of atomic model building. I was wondering if the authors could try to make a bit more efforts to improve the resolution. For example, I realized that the authors used C8 to refine the final map and stopped there. I felt that based on the C8-symmetrized map, a symmetry expansion, followed by focused refinement and classification using C1, and re-refinement of the best classes might help. Also, just out of intuition, I felt that most cryo-EM programs dealing with a structure like this filament would confuse the chirality of the structure, leading to an average of the correct and Z-flipped maps. This is just my guess. There is a very simple way to evaluate if this was the case: perform two independent ab initial refinements (ideally from the very beginning, which avoid inter-dependence of the references used) and compare the final Euler angle distributions of individual particles from the two independent refinements. In the ideal case, every single particle will converge to the same Euler angles, and in the worst case, particles are just randomly classified into the correct and false hands.

We thank reviewer for the insightful suggestion. We appreciate reviewer's concern regarding the potential chirality issue in our cryo-EM analysis. We agree that, a structure like filament

could be prone to chirality ambiguities especially when averaging the maps that correspond to different handedness. This is particularly true if that the sample conformation is homogeneous. However, in our case where sample is conformationally heterogeneous, the filaments are mixed with all various conformations (e.g. bended, straight, stretched), the Ab-initio results using the particles from the very beginning will vary from run-to-run. These variations are more due to the conformational heterogeneity than handedness as indicated by our 2D and 3D classification results. After separating the different conformations, we obtained a relatively homogenous conformation as shown in our final map. To evaluate whether the particles we used for our final map have chirality issue, we performed another two independent Ab-initio refinements. The Euler angle distributions converged closely to each other, indicating that the structure is determined with the correct chirality. We've included a detailed description of this validation process in the revised manuscript to clarify our approach and the results.

We also thank reviewer for the suggestion '*I felt that based on the C8-symmetrized map, a symmetry expansion, followed by focused refinement and classification using C1, and re-refinement of the best classes might help*'. We followed reviewer's suggestion by implementing symmetry expansion and classification to select the best class for reconstruction. The resulting map was reported with the resolution of 3.9 Å, seeming to have a nominal improvement. However, there was no significant improvement in side-chain appearance and the map became more fragmented losing the connectivity of the alpha-helix. We therefore chose not to update the map. We nevertheless appreciate reviewer's suggestion for their willingness of help.

Other comments:

Line 26: "the underlying molecular mechanism have remained unclear..." → should be singular form, "mechanism has remained unclear."

Thank you. The error is corrected.

Line 35: "...significant reduction in embryonic viability and the formation of multipolar spindles..." No major issue and I did understand what the authors intended to say here, but I felt the sentence can be a bit confusing for general readership because I could also interpret this as "reduction in both A (embryonic viability) and B (the formation of multipolar spindles)". Something like "reduction in embryonic viability, along with the formation of multipolar spindles" may resolve this confusion.

The error is corrected.

Line 214: "assembly of these variants were assessed using dynamic light scattering (DLS)", should be 'was assessed'.

The error is corrected.

Line 367: "Accession Codes" → It seems "Data Availability" is the standard term in Nature press.

The error is corrected.

Reviewer #2 (Remarks to the Author):

This study by Agostini et al. dissected the structure of microtubule-associated protein SSNA1 through cryo-EM, uncovering key residues (or regions) for effective self-assembly of SSNA1 fibril, microtubule binding and remodeling, and centriole functionality. Initially, they generated ssna-1 knockout nematodes and observed significant embryonic lethality, underscoring its important physiological functionality. They purified C. elegans SSNA-1 (3E) mutants for cryo-EM-based structure determination, and found SSNA-1 forms double-stranded coiled-coils where intra-filament junctions were triple-stranded. Through analysis of truncated SSNA1 mutants, they identified several key residues such as Y15, R18 and Y97, which are crucial for long fibril formation and microtubule binding. Remarkably, these residues (or SSNA1 fibril formation) are also indispensable for in vitro microtubule branching induced by SSNA1. Finally, this study demonstrated that fibril formation and microtubule binding activity of SSNA-1 are imperative for its physiological roles including centriole maintenance.

These results offer significant advances in our understanding of the working mechanism of SSNA1 in centriole functionality and microtubule branching by providing structural and molecular insights. However, there are some critical issues remaining to be addressed before publication.

Major comments:

1) The authors identified an SSNA-1 mutant with 3 point mutations (R18E/R20E/Q98E or 3E) for further structure determination, as wild-type C. elegans SSNA-1 formed bundled filaments that were not suitable for analysis. However, they also found that several residues, including R18, are crucial for the functionality of SSNA1 including microtubule binding. This indicates that SSNA-1 (3E) may not fully represent the physiological state of SSNA1 in vivo. Does ssna-1 (3E) mutant animals exhibit significant embryo lethality as ssna-1 KO animals? At least, how the physiological functionality of SSNA-1 (3E) mutant diverge from wild-type SSNA-1 should be discussed.

We agree with the reviewer's excellent point that the R18E mutation and the lack of bundling, while feasible for structural analysis, is likely to cause physiological disturbances. To address this point, we have created the *c.elegans* SSNA-1 (3E) mutant and quantified its viability. The 3E mutant has an average viability of 32%, similar to the knockout (added to Fig. 6). We nevertheless think the essential structural mechanism of how the fibril is connected is retained, as consistently shown with alpha-fold prediction (now added to Fig. S3) as well as the periodical features that are displayed in the FL (Fig. 4C). The reviewer is correct that we have a limited understanding of the ability of WT SSNA-1 to make a thicker bundle and the relevance of it. We also speculate that, based on the enigmatic localizations of SSNA1 at cilia, basal body, and at the centrosome satellite, there is a possibility that SSNA1 may have multiple localizations with overlapping but distinct functions and interaction partners. Curious observation of *c.elegans* R18E mutant that loses the localization at the satellite might indicate that the bundling of SSNA1 is important for the recognition of centriole and its stabilization but the microtubule binding may be a secondary function for centriole recognition.

Together with the added analysis in Fig. 6 and Fig. S3, the discussion about the limitation of our structural insights and the necessity of understanding the functional relevance of the SSNA-1 bundling in the future is now added in discussion.

2) Line 233. "The modifications R18E/Y97E, Y15E/Y97E, and Y97E, disrupted long fibril formation observed for the full-length, and instead thin, shorter fibrils were observed (Fig. 4A-B, 4I-K)."

However, in Figure 4B and 4J, it appears that Y15E/Y97E still forms long fibrils as wild-type SSNA1.

We thank the reviewer for your careful observation. We investigated the original data and we realized that the batch of the protein used contained small aggregates. We repeated the analysis and now the data shows the consistent indication as negative stain. The figure 4B and 4J are updated.

3) Previously, the authors' group identified that SSNA1 directly induced microtubule branching in vitro, although a subsequent study by another group failed to observe SSNA1-induced MT branching (Lawrence et al., 2021, eLife). In this study, the authors illustrated that 4.29% (7 of 163) of MTs were branched induced by FL SSNA-1. Furthermore, SSNA-1 mutants those failed to form long fibrils show deficiency in inducing MT branching. Considering the relatively low branching frequency, I recommend the authors analyze a larger number of MTs in each group for statistics (e.g., 500-1000 MTs per group). This would make their conclusion more authentic.

To ensure the statistical robustness of our findings, we expanded our dataset and analyzed over 1,000 microtubules, providing a more reliable assessment of the observed branching events.

Minor comments:

1) Line 31: “..., highlighting that self-assembly of SSNA-1 facilitates effective microtubule interaction by creating hubs along a fibril.”

This conclusion is not directly supported by their results.

We have changed the phrase, from ‘highlighting’ to ‘suggesting’.

2) *MT branching assay is missing in Methods section.*

The branching assay was described as co-polymerization assay. We have now made a new section specifying the branching assay in Methods section.

3) Line 147: “we purified *C. elegans* SSNA-1 and...”. *It would be better to specify that SSNA-1 was purified from bacteria.*

We thank the reviewer for pointing this out. Now it reads ‘we recombinantly prepared *C. elegans* SSNA-1 using *E.coli*’

4) Line 329: “...it is a critical member of the centriole coiled coil fibrillar protein with its key role in maintaining centriole stability”. *However, no evidence in this study shows that SSNA-1 is critical for centriole stability. Would it be more accurate to replace ‘stability’ with ‘functionality’?*

We agree. It is rephrased to ‘functionality’.

5) Line 439: “for 5’ ” *Does this mean “for 5 min”?*

We have fixed the error.

6) *The authors deposited too many conclusions into figure legends. For example: Line 616: “The results indicate that the formation of fibrils is not essential for the binding of SSNA-1 to microtubules, however, the SSNA-1 N-terminus, including residue R18 and its neighbouring residues, is critical for microtubule-binding.”*

*Line 628: “These results indicate that microtubule branching requires SSNA-1 to bind microtubules and form oligomeric or fibrillar assemblies.” Line 638: “These results show that residues Y15, R18, Y97, which are important for fibril formation, are also critical for the viability of *C. elegans*.”*

These conclusions should be removed from figure legends.

We corrected the legend portion where too much scientific conclusions are described.

Reviewer #3 (Remarks to the Author):

*In this manuscript Agostini et al. investigate the self-assembly properties of *C. Elegans**

*SSNA-1 protein that associates with microtubules and is implicated in the structural stability of centrosomes (accompanying paper). Using mutated SSNA-1 constructs, the authors report that mutant SSNA-1 forms anti-parallel coiled coils which then assemble into higher-order fibrils forming specific triple-stranded helical junctions. The authors find that these triple-stranded junctions are also sites of microtubule interaction. Furthermore, the authors investigate the ability of several mutant and deletion SSNA-1 constructs to form microtubule branches, previously reported for wt *C. reinhardtii* SSNA-1 homolog. Overall, the manuscript provides little substantive insight into the mechanism by which SSNA-1 supports centriole structure – a function reported in the accompanying paper. As such, the limited results of the present manuscript could be fully absorbed into the other SSNA-1 paper by the same group of authors.*

We have carefully considered the possibility of combining our two papers into one, discussing it among the authors and with the editor. Respectfully, we believe that merging the two papers would not be feasible. It would be extremely challenging to adequately address both the structural and biological data in a single manuscript without making it excessively lengthy and difficult to navigate. We believe that keeping the papers separate allows for a more focused and thorough presentation of each aspect, ultimately benefiting the readers and the clarity of the work. We nevertheless thank the reviewer for your suggestion.

Specific comments:

Figure 1: New results in this paper originate from the subsequent structural studies; as such, current Figure 1 does not fit into the present manuscript. Rather, these results are covered in the accompanying manuscript, where they seem to be more appropriate.

We believe it is essential to highlight the biological impact of SSNA1 to provide readers with an introduction to its significance. As this forms an integral part of our study, we have included relevant biological data in this manuscript. Given the back-to-back nature of this and the accompanying paper, there is some intentional conceptual overlap in the information presented. We feel this redundancy is necessary to ensure that each paper can be read independently without requiring readers to constantly refer to the other. Our aim is to provide sufficient context and clarity within each manuscript to enhance accessibility and understanding for a broader audience.

Figure 2. Given the significant mutations in the protein that were induced in order to determine the fibril structure, it is unclear how physiologically relevant is this 8-filament tube structure? Notably, the exact residues that were mutated here are implicated in fibril formation. Thus the reported structure may well be an artifact of the induced mutations. Furthermore, how/where would this fit on the microtubule and within the centriole? Note the apparent discrepancy from the previously reported structure by Basnet et al. – this should be addressed.

We agree with the reviewer's excellent point that the R18E mutation and the lack of bundling, while feasible for structural analysis, is likely to cause physiological disturbances. To address this point, we have created the *c.elegans* SSNA-1 (3E) mutant and quantified its viability. The 3E mutant has an average viability of 32%, similar to the knockout (now added to Fig. 6A).

We nevertheless think the essential structural mechanism of how the fibril is connected is retained, as consistently shown with alpha-fold prediction (Fig. S3) as well as the periodical features that are displayed in the FL (Fig. 4C). However, the reviewer is correct that we have a limited understanding of the ability of WT SSNA-1 to make a thicker bundle and the relevance of it. We also speculate that, based on the enigmatic localizations of SSNA1 at cilia, basal body, and at the centrosome satellite, there is a possibility that SSNA1 may have multiple localizations with overlapping but distinct functions and interaction partners. Curious observation of *c.elegans* R18E mutant that loses the localization at the satellite might indicate that the bundling of SSNA1 is important for the recognition of centriole and its stabilization but the microtubule binding may be a secondary function for centriole recognition. Together with the new analyses added in Fig. 6A and Fig. S3, the discussion about the limitation of our structural insights and the necessity of understanding the functional relevance of the SSNA-1 bundling in the future is now added in discussion.

In our previous study (Basnet et al., 2018), we did not resolve the SSNA-1 structure to determine whether its coiled-coil conformation was parallel or anti-parallel. At the time, the structure of SSNA-1 was unavailable, and we followed the conventional assumption - based on prior publications, that the coiled-coil runs in parallel by citing other research. We recognize that this ambiguity may have been misleading. To rectify this, we have explicitly addressed this point in the current manuscript, clarifying the structural orientation of SSNA-1 based on our new data.

Figure 3. This is a fully alphafold figure which should be incorporated within or as a supplemental to Figure 2. Also, the structure appears contradictory to previously published structure by the authors (reporting parallel, as opposed to antiparallel dimers). Furthermore, the authors note that R18 is a key residue for the triple helical junction but they mutated it for their structural determination. If this is a key residue then it would be helpful to have a structure that actually reflects this. Can the authors predict what the structure would look like without induced mutations?

We apologize for confusion caused by this figure. The ribbon representation in Fig. 3 is a structure using an AlphaFold prediction as an initial starting point for refinement. AlphaFold prediction was fitted to the cryo-EM map, then performed rigid body refinement in PHENIX, and real-space refinement in COOT and PHENIX. Therefore, this representation is not solely a prediction but it is refined based on the cryo-EM structure. While we chose not to include side chains in the model building due to the limited resolution, we believe this figure provides insights into how the triple-stranded coiled-coils connect fibrils. To address this, we have added a statement that this is an actual refined structure in the figure legend for clarity.

Furthermore, we performed AlphaFold predictions on wildtype SSNA-1, which confirmed that its predicted structural organization is identical to that of the 3E mutant reported in this study. This shows that the 3E mutation does not alter the overall structural architecture of SSNA-1. We believe that the primary effect of the 3E mutation is to alter the charge distribution, which in turn likely modifies the interaction dynamics between SSNA-1 bundles, rather than fundamentally changing their structural conformation, and that facilitated our structural analysis. We added the wildtype prediction in new Fig. S3, and added corresponding statement to text. Regarding the response about the description of parallel SSNA-1 in the previous paper, please see above.

Figure 4. Analysis of the 3E mutant should be included here. Also, the subsequent categorization of fibril formation should be done based on these DLC results. Can the spectra be quantified?

The 3E data has been moved from supplementary to Fig. 4. The mass analysis was quantified and the ratio of smaller peak (in percentage) is added to the figure (note: bigger peak (%) = 100% – smaller peak (%)).

Figure 5. This is a particularly weak figure as it reports extremely rare occurrences of what the authors report as microtubule branching and correlate this with the potential for fibril formation. The statistics are very poor, the images of branching tenuous at best, and the finding that these events are so rare puts in question the physiological relevance of these observations.

To ensure the statistical robustness of our findings, we expanded our dataset and analyzed over 1,000 microtubules, providing a more comprehensive assessment of the observed branching events. However, the biological significance of this branching remains an open question. At this stage, we cannot determine whether it represents a physiological process, a purely biophysical phenomenon, or a pathological occurrence linked to cytoskeletal dysregulation. While these observations are intriguing, we do not claim that they reflect a naturally occurring mechanism in vivo. We are presenting this finding as a structural phenomenon that merits further exploration.

Figure 6. This figure fails to provide a cohesive picture of the results. It is completely unclear how the structures presented in Figures 2 & 3 fit into this schematic? The binding of SSNA1 fibrils to microtubules as well as within centrioles has not been clarified by the presented data.

We would like to clarify that our study establishes a correlation between fibril formation, microtubule binding at amino acid level, derived by the structural information in Fig. 2 and 3, and its localization at the centriole and relevance to the centriole integrity and cell viability. In addition, the detailed analysis is available in the accompanying paper. To better clarify

and to indicate the availability of detailed analysis, we added the statement and the reference of the accompanying paper at the figure legend.